# The Geriatric Nutritional Risk Index Is an Optimal Evaluation Parameter for Predicting Mortality in Adult All Ages Hemodialysis Patients: A Korean Population-Based Study

**DOI:** 10.3390/nu15173831

**Published:** 2023-09-01

**Authors:** Do Hyoung Kim, Young-Ki Lee, Hayne Cho Park, Bo Yeon Kim, Miri Lee, Gui Ok Kim, Jinseog Kim, Ajin Cho

**Affiliations:** 1Department of Internal Medicine, Kangnam Sacred Heart Hospital, Hallym University College of Medicine, Seoul 07441, Republic of Korea; dhkim6489@hallym.or.kr (D.H.K.); km2071@hallym.or.kr (Y.-K.L.); haynepark798@hallym.or.kr (H.C.P.); 2Healthcare Review and Assessment Committee, Health Insurance Review and Assessment Service, Wonju 26465, Republic of Korea; kimby01@hira.or.kr; 3Quality Assessment Division 1, Quality Assessment Department, Health Insurance Review and Assessment Service, Wonju 26465, Republic of Korea; mmmm22@hira.or.kr; 4Quality Assessment Management Division, Quality Assessment Department, Health Insurance Review and Assessment Service, Wonju 26465, Republic of Korea; rrnlfl52@gmail.com; 5Department of Big Data and Applied Statistics, Dongguk University, Gyeongju 13557, Republic of Korea; jinseog.kim@gmail.com

**Keywords:** Geriatric Nutritional Risk Index, cutoff value, hemodialysis, mortality, end-stage renal disease

## Abstract

The Geriatric Nutritional Risk Index (GNRI) is a nutritional screening tool used for predicting mortality in patients undergoing hemodialysis (HD). This study investigated the cutoff values for the GNRI for predicting mortality in HD patients using Korean HD quality assessment data from 2015. To identify the optimal GNRI cutoff value, we used Harrell’s C-index with multivariate Cox regression models. The highest value of C-index was identified as the cutoff value of GNRI for all-cause mortality in this population. In total, 34,933 patients were included; 90.8 of GNRI was the highest value of C-index, and it was used as a cutoff value to predict mortality; 3311 patients (9.5%) had GNRI values < 90.8, and there were 12,499 deaths during the study period. The mean follow-up period was 53.7 months. The crude mortality rates in patients with GNRI values < 90.8 and ≥ 90.8 were 160.4/1000 and 73.2/1000 person-years respectively. In the fully adjusted Cox model, patients with a GNRI < 90.8 had a 1.78 times higher risk of mortality than those with a GNRI ≥ 90.8. These findings suggest that the optimal GNRI cutoff value is 90.8 for predicting mortality in maintenance HD patients.

## 1. Introduction

Malnutrition is a common problem and is associated with high mortality in patients with end-stage renal disease (ESRD) undergoing hemodialysis (HD) [1,2,3]. Several nutritional screening tools have been developed and adapted for the nutritional assessment of HD patients. The Subjective Global Assessment (SGA) method and Malnutrition–Inflammation Score (MIS) are validated tools for this purpose in HD patients. However, these methods require a subjective assessment [4,5,6]. The Geriatric Nutritional Risk Index (GNRI), which is calculated from height, weight, and serum albumin concentration, is a simple method for assessing nutritional status and is used widely in various clinical situations [7]. The GNRI was developed primarily for elderly people and is known to be useful for predicting poor outcomes [8].

A lower GNRI value indicates a higher risk of malnutrition. A previous study validated the use of the GNRI to identify nutritional risk and reported that a GNRI value < 91.2 indicated a high possibility of malnutrition in patients undergoing HD [9]. In patients with acute heart failure, a GNRI value < 92 on admission was found to be independently associated with worse clinical outcomes, and the GNRI was superior to other measures of nutritional risk [10]. Previous studies have reported that the GNRI can be used to predict mortality in patients with ESRD [11,12,13]. In a previous study, we found that patients with a GNRI value < 97.7 had increased mortality compared with those with a value ≥ 97.7 [14]. However, few studies have investigated the optimal cutoff value for predicting poor outcomes in patients with ESRD undergoing HD. The suggested GNRI values were different among various clinical settings, and a small number of the subjects were included in previous studies. The nutritional status of patients with ESRD constantly changes. Therefore, prognostic values of GNRI to predict mortality might change over time. However, relationships between GNRI changes and mortality remain unclear in patients undergoing HD.

Even with technical advances in dialysis, the mortality risk is higher in patients undergoing HD than in the general population [15]. Understanding the prognostic factors and instituting timely interventions contribute to better clinical outcomes in this population. Using national representative cohort data, the present study examined the optimal cutoff value of the GNRI for predicting mortality in patients undergoing HD. We also investigated associations between mortality and GNRI value as a time-varying factor in this population.

## 2. Materials and Methods

### 2.1. Study Population

The National Health Insurance System (NHIS) in Korea is a single-payer system to which almost all patients undergoing HD belong. The HIRA (Health Insurance Review and Assessment Service) is a government organization that assesses medical services and maintains quality standards by reviewing healthcare claims. Medical providers are required to submit all inpatient and outpatient claims to HIRA for reimbursement of medical procedure costs covered by NHIS. HIRA assesses the quality of HD in each dialysis center under specified criteria, including structural, procedural, and outcome domains, every 3 years. We have previously published the details of the HD quality assessment [16].

In the current study, we enrolled a total of 34,950 HD patients who were included in the HD quality assessment data in 2015 (Figure 1). Adults aged ≥18 years who had received conventional maintenance HD at least twice were included. Patients with missing values for body weight (*n* = 10) or albumin concentration (*n* = 7) were excluded from the analysis to identify the optimal cutoff GNRI value. Patients who had GNRI values in 2015 and 2018 (*n* = 16,896) were included in the analysis of the association between the GNRI as a time-varying factor and all-cause mortality. The study was conducted in accordance with the Declaration of Helsinki and was approved by the Institutional Review Board (IRB) of the Kangnam Sacred Heart Hospital, Seoul, Republic of Korea (HKS-2021-11-043). The IRB waived the need for informed consent because the study participants were deidentified.

### 2.2. Assessment of the GNRI

The GNRI is calculated using the following equation [8].
GNRI = (14.89 × albumin (g/dL)) + (41.7 × (body weight/ideal body weight))

The ideal weight was calculated using height and a body mass index (BMI) of 22 kg/m^2^ [17]. If a patient’s body weight was above the ideal body weight, the ratio of body weight to ideal body weight was replaced with 1.

### 2.3. Data Collection and Measurement

HD quality assessment requires data from dialysis facilities collected using a web-based data collection system [18]. The HD service providers enter the general information regarding HD treatments and facilities. Patient factors, including age, sex, primary cause of ESRD, dialysis vintage, single-pool Kt/V, BMI, pre-dialysis systolic and diastolic blood pressures, and laboratory findings (serum hemoglobin, serum albumin, calcium, and phosphorus concentrations) were collected at the time of enrollment and used in this study. Comorbid conditions were identified from the International Classification of Diseases-10 codes in the NHIS claim data from January to December 2015. Comorbidities included ischemic heart disease (I20–I25), congestive heart failure (I50), cerebrovascular disease (I60–I64, I69), diabetes (E10–E14), hypertension (I10–I13, I15), and atrial fibrillation (I48).

The primary outcome was all-cause mortality, which was regarded as death if a patient’s data were extracted from enrollment in the HIRA. Patients who received a kidney transplant during a follow-up period or who followed until the end of the study (30 November 2021) were censored.

### 2.4. Statistical Analysis

Differences in patient characteristics according to the GNRI value were assessed using quartiles and one-way analysis of variance for continuous variables and the chi-squared test for categorical variables. Cox proportional hazard regression analysis was used to assess the associations between all-cause mortality and GNRI values. Three different multivariate models were used after adjustment, and a total of 15 variables were used in the full model. Model 1 included age, sex, dialysis vintage, and GNRI value. Model 2 included the variables in Model 1 plus comorbid conditions. Model 3 (full model) included all previous variables plus laboratory findings (hemoglobin, calcium, and phosphorus concentrations) and Kt/V.

To identify the GNRI cutoff value for mortality prediction, we calculated Harrell’s C-index of GNRI in the multivariate Cox regression model (Model 3) [19]. The highest value of C-index was identified as the cutoff value of GNRI for all-cause mortality in this population. Sensitivity, specificity, positive/negative likelihood ratio, and positive/negative predictive values with the cutoff value were calculated using receiver-operating characteristic curve analyses. Distribution and cutoff points of GNRI for all-cause mortality were presented in different age and sex groups.

The mortality risk of patients with less than the GNRI cutoff value was evaluated using Cox models. Patient survival was described using the Kaplan–Meier method and compared between groups using the log-rank test. Time-varying Cox regression models were performed to evaluate the mortality risk of a low GNRI as a time-varying factor in 2015 and 2018. Age, dialysis vintage, and laboratory data were considered time-varying factors in these models. All *p* values were two-sided, and *p* < 0.05 was considered to be significant. Statistical analyses were performed using R version 4.0.5 (R Foundation for Statistical Computing, Vienna, Austria. URL https://www.R-project.org/ accessed on 31 May 2021).

## 3. Results

### 3.1. Baseline Characteristics of the Patients

The baseline characteristics of the patients grouped by GNRI quartiles are shown in Table 1. The overall mean age was 60.2 years, and 20,534 (58.8%) were men. The mean dialysis vintage was 5.6 years; 21,486 (61.5%) had diabetes, and 29,654 (84.9%) had hypertension. The mean GNRI value was 98.7, the average age decreased, and the percentage of men increased as the GNRI quartile increased. The percentages of patients with ischemic heart disease, heart failure, cerebrovascular disease, or atrial fibrillation also decreased as the GNRI quartile increased. Serum phosphate concentration increased as the GNRI quartile increased.

### 3.2. The GNRI Cutoff Value

The C-index of GNRI is shown in Figure 2; 90.8 of GNRI was the highest C-index value, and the value indicated the optimal GNRI cutoff for mortality prediction.

The distribution of the GNRI values and the adjusted hazard ratios (HRs) for the GNRI used in the three multivariate Cox models are shown in Figure 3. The mortality risk ratio steeply increases when the GNRI value is less than 90.8.

Sensitivity, specificity, positive/negative likelihood ratio, and positive/negative predictive values of the cutoff value were 0.85, 0.06, 0.90, 2.52, 0.33, and 0.42, respectively; 3311 patients (9.5%) had GNRI values < 90.8 in 2015. Patients with a GNRI value < 90.8 were older and had lower serum concentrations of hemoglobin, calcium, and phosphate.

### 3.3. Cutoff Values for All-Cause Mortality in Different Age and Sex Groups

Figure 4 shows the C-index of GNRI in the multivariate Cox model (Model 3) according to age and sex groups. The suggestive GNRI cutoff value in ages < 65, 65–74, and ≥75 were 96.9, 92.0, and 89.9 respectively. The values for women and men were 94.9 and 89.5.

The distribution of the GNRI values and the adjusted HRs for all-cause mortality are shown in Figure 5. The mortality risk ratio steeply increases when the GNRI value is less than the cutoff values in all the groups.

### 3.4. The Cutoff Values and All-Case Mortality

There were 12,499 deaths, and 3512 received kidney transplantation during the study period out of patients in the 2015 cohort. The mean follow-up period was 53.7 months. The crude mortality rates in patients with a GNRI value < 90.8 and ≥90.8 were 160.4/1000 and 73.2/1000 person-years, respectively (Table 2). Crude mortality rates by subgroups demonstrated that patients with GNRI less than the cutoff value in each group have significantly higher mortality rates than the others.

The survival curves in the patients with GNRI < 90.8 and ≥90.8 were significantly different (log-rank *p* < 0.001) (Figure 6).

In the fully adjusted Cox model, patients with a GNRI value < 90.8 had a 1.78 times higher mortality risk compared with those with a GNRI value ≥ 90.8 (Table 3). In each age subgroup, relative mortality risks in patients with GNRI less than the cutoff value were significantly high compared with the others. Male and female patients with GNRI less than the cutoff value have 1.92 and 1.59 times higher mortality risk, respectively.

### 3.5. Associations between All-Cause Mortality and the GNRI as a Time-Varying Factor

GNRI values were available for 16,896 patients in 2015 and 2018. In these patients, Cox regression analyses for all-cause mortality were performed with a GNRI value as a time-varying factor. A GNRI value was considered a continuous or categorical variable in the Cox models (Table 4). In all subgroup patients, a GNRI less than the cutoff value was associated with higher mortality during the follow-up.

## 4. Discussion

In this study, we identified the GNRI cutoff value (90.8) for predicting all-cause mortality in patients with ESRD undergoing HD. During the follow-up period, a decreased GNRI value was associated with an increased risk of mortality. Assessment of the mortality risk showed that a GNRI value < 90.8 was associated with a 1.78 times higher relative risk of mortality than those with a GNRI ≥ 90.8. The optimal cutoff value of GNRI was different by age and sex groups. These results provide new information about the use of the GNRI cutoff value for predicting mortality and the possible benefits of maintaining the GNRI for clinical outcomes.

The GNRI was developed as a nutritional screening tool primarily for elderly patients. Yamada et al., first investigated the usefulness of the GNRI compared with other nutritional assessment methods such as the MIS, Mini Nutritional Assessment-Short Form, Nutritional Risk Score, Malnutrition Universal Screening Tool, and Malnutrition Screening Tool, and found the GNRI to be a good indicator of nutritional status in patients undergoing HD [9]. They reported a GNRI cutoff value < 91.2 as the most accurate for identifying malnourished patients, according to the MIS. Another study at a university hospital reported an optimal cutoff GNRI value of < 96 for nutritional screening among older patients [19]. The suggested GNRI values for identifying malnutrition may differ between various clinical settings. In addition to its use as a tool to assess nutritional status, the GNRI also appears to be a good predictor of mortality in patients undergoing HD.

Only a few studies have validated the optimal GNRI cutoff value for predicting overall survival in ESRD patients. In a study of 104 patients aged > 65 years undergoing HD, a lower GNRI was associated with a higher risk of overall mortality in the elderly ESRD patients undergoing maintenance HD, and a GNRI value < 92 was suggested as the cutoff for predicting mortality [20]. Predictions using the categorization of serum concentrations to identify risk, as in previous studies, are more difficult to interpret when comparing cutoff values because of the heterogeneous reference values used in each study. Some studies have used a GNRI cutoff value of < 92 [12,20], but these included a small number of ESRD patients. A larger study may be useful for further defining the prognostic value of the GNRI and suggesting a reference value for predicting mortality. We used nationwide cohort data and determined the cutoff value using multivariate Cox proportional hazard models instead of arbitrary categories. The cutoff value obtained here may provide a reference for identifying the highest mortality risk in ESRD patients. However, this study included a large population of patients; further validation studies are needed to confirm the cutoff value reported here.

The present study also analyzed the association between a time-varying GNRI value and mortality. Our previous study showed that a negative GNRI trend was associated with an increased risk of all-cause mortality in patients with incident HD [14]. A previous study that included 119 patients undergoing HD also reported lower all-cause and cardiovascular survival rates in groups with a lower baseline GNRI and decreased GNRI [12]. Beberashili et al., reported that longitudinal changes in the GNRI value over time were associated with appropriate changes in the biomarkers of nutrition, inflammation, and body composition parameters [21]. Our study showed that having GNRI less than the cutoff value during the follow-up period is a significant prognostic factor for mortality in HD patients. Longitudinal assessment of GNRI may be useful for identifying risk groups for mortality and the appropriate treatment.

Our study has some limitations. First, we cannot exclude the possibility of residual confounding factors, although most factors were adjusted for. Second, this was a retrospective cohort study, which cannot be used to prove causality and further validation studies are needed to confirm the associations. Third, this study included only Korean ESRD patients, and the ability to generalize to other populations may be limited. Fourth, the patients who were admitted to the hospital during the HD quality assessment were excluded. However, the GNRI values were obtained when the patients were in a stable condition that did not require hospitalization. Fifth, we used only two GNRIs for each patient to assess the change with time. Finally, the serum albumin concentration may be affected by various clinical conditions, such as infection, but we could not obtain information about this.

In conclusion, the present study showed that the cutoff value of GNRI for mortality was 90.8 in patients undergoing HD. A GNRI value < 90.8 and persistently having a GNRI less than the cutoff value was associated with the highest relative risk of mortality. To our knowledge, this is the first study to use a large population-based database to identify the optimal GNRI cutoff value for predicting mortality. Although further validation studies of this value are needed, we suggest that a GNRI value < 90.8 is a simple and useful prognostic factor in patients with ESRD undergoing HD.

## Figures and Tables

**Figure 1 nutrients-15-03831-f001:**
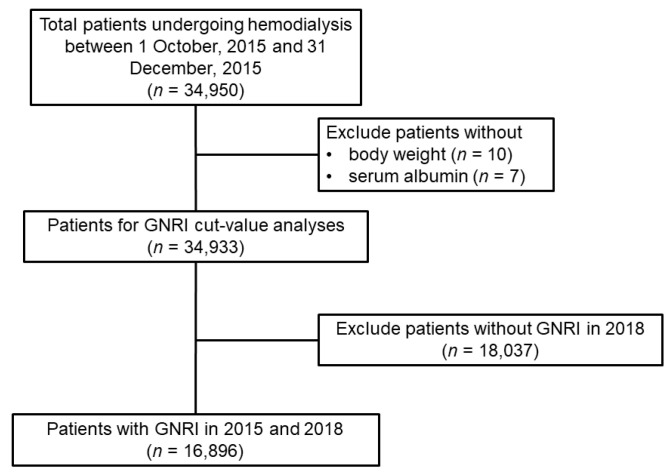
Flow diagram of the study population.

**Figure 2 nutrients-15-03831-f002:**
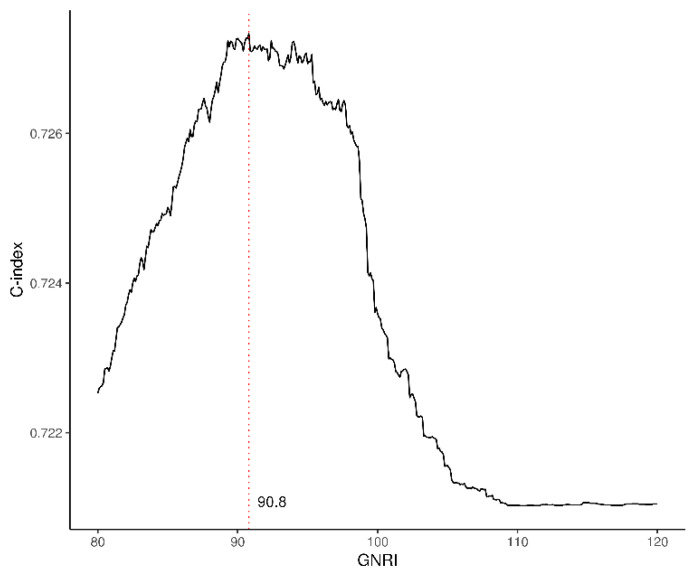
The C-index of GNRI. The red dot line indicated a GNRI value with the highest C-index.

**Figure 3 nutrients-15-03831-f003:**
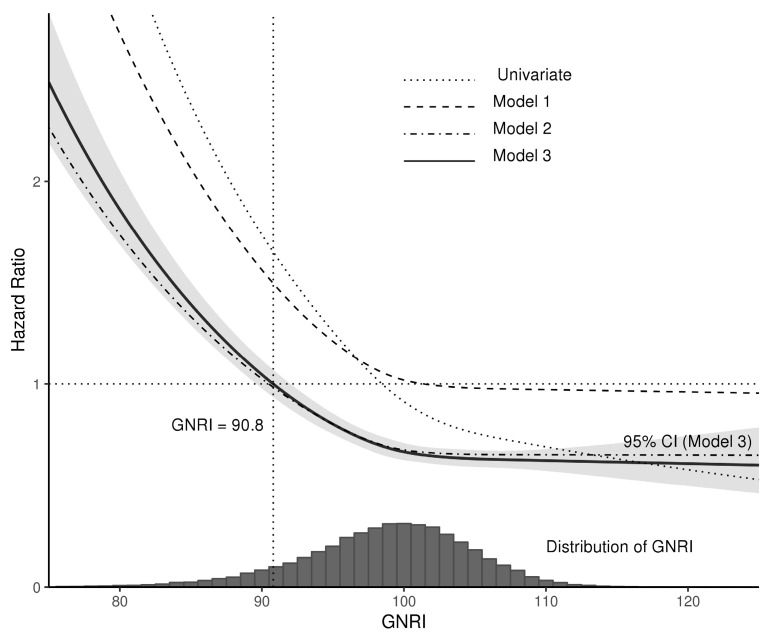
The relative risk of mortality [HR (95% CI)] with GNRI. Multivariate Model 1 included age, sex, dialysis vintage, and GNRI; Model 2 included the variables of Model 1 plus comorbid conditions; Model 3 (full model) included all previous variables plus laboratory findings.

**Figure 4 nutrients-15-03831-f004:**
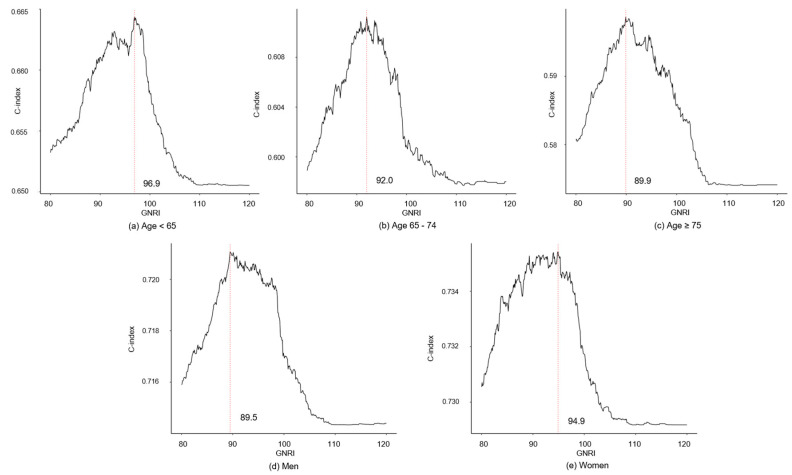
The C-index of GNRI in subgroups. The red dot line indicated a GNRI value with the highest C-index.

**Figure 5 nutrients-15-03831-f005:**
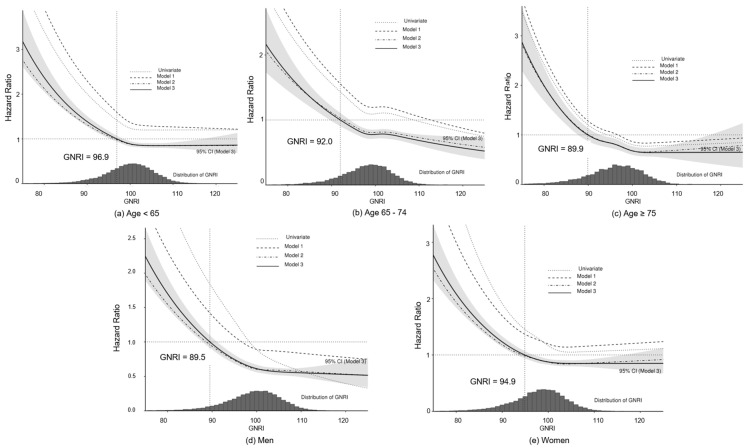
Relative risk of mortality with GNRI in subgroups. Multivariate Model 1 included age, sex, dialysis vintage, and GNRI; Model 2 included the variables of Model 1 plus comorbid conditions; Model 3 (full model) included all previous variables plus laboratory findings.

**Figure 6 nutrients-15-03831-f006:**
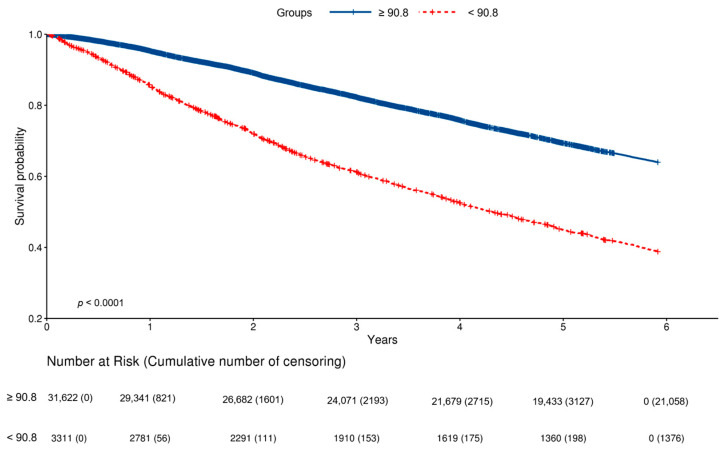
Survival curves in patients with GNRI < 90.8 vs. GNRI ≥ 90.8.

**Table 1 nutrients-15-03831-t001:** Baseline characteristics by quartiles of patients with GNRI in 2015.

	Total (*n* = 34,933)	Quartile 1 62.4–95.2 (*n* = 8917)	Quartile 2 95.3–99.2 (*n* = 8426)	Quartile 3 99.3–102.7 (*n* = 9284)	Quartile 4 102.8–153.9 (*n* = 8306)	*p*-Value
Age, year	60.2 ± 13.0	64.2 ± 12.9	61.2 ± 12.4	59.4 ± 12.2	55.8 ± 12.3	<0.001
Men	20,534 (58.8)	4797 (53.8)	4554 (54.0)	5429 (58.5)	5754 (69.3)	<0.001
Dialysis vintage, year	5.6 ± 5.1	5.8 ± 5.5	5.8 ± 5.3	5.5 ± 4.9	5.3 ± 4.6	<0.001
SBP (mmHg)	141.2 ± 15.6	140.0 ± 16.1	140.8 ± 15.3	141.3 ± 15.3	142.9 ± 15.3	<0.001
DBP (mmHg)	77.6 ± 9.6	75.9 ± 10.0	77.2 ± 9.5	77.9 ± 9.3	79.3 ± 9.1	<0.001
Body mass index (kg/m^2^)	22.4 ± 3.4	20.2 ± 3.1	22.1 ± 3.2	23.4 ± 3.1	23.8 ± 2.9	<0.001
Kt/V	1.6 ± 0.3	1.6 ± 0.3	1.6 ± 0.3	1.5 ± 0.3	1.5 ± 0.3	<0.001
Comorbid conditions						
Diabetes mellitus	21,486 (61.5)	5564 (62.4)	5170 (61.4)	5751 (61.9)	5001 (60.2)	0.021
Hypertension	29,654 (84.9)	7682 (86.1)	7180 (85.2)	7845 (84.5)	6947 (83.6)	<0.001
Ischemic heart disease	12,088 (34.6)	3328 (37.3)	3036 (36.0)	3148 (33.9)	2576 (31.0)	<0.001
Cerebrovascular disease	3116 (8.9)	1016 (11.4)	763 (9.1)	745 (8.0)	592 (7.1)	<0.001
Heart failure	5123 (14.7)	1540 (17.3)	1286 (15.3)	1293 (13.9)	1004 (12.1)	<0.001
Arrhythmia	1849 (5.3)	640 (7.2)	466 (5.5)	464 (5.0)	279 (3.4)	<0.001
Hemoglobin, g/dL	10.7 ± 0.9	10.6 ± 0.9	10.7 ± 0.8	10.8 ± 0.8	10.8 ± 0.9	<0.001
Albumin, g/dL	4.0 ± 0.4	3.6 ± 0.3	3.9 ± 0.2	4.1 ± 0.2	4.4 ± 0.2	<0.001
Calcium, mg/dL	9.0 ± 0.8	8.8 ± 0.8	9.0 ± 0.8	9.0 ± 0.8	9.2 ± 0.8	<0.001
Phosphorus, mg/dL	4.9 ± 1.3	4.6 ± 1.3	4.9 ± 1.3	5.1 ± 1.3	5.3 ± 1.4	<0.001

Data are number (percentage) and mean ± standard deviation. Abbreviations: SBP, systolic blood pressure; DBP, diastolic blood pressure; GNRI, geriatric nutrition risk index.

**Table 2 nutrients-15-03831-t002:** Crude mortality of patients with GNRI in 2015.

	Total *n* = 34,933		Age < 65 *n* = 21,324		Age 65–74 *n* = 8706		Age ≥ 75 *n* = 4903		Men *n* = 20,534		Women *n* = 14,399	
	GNRI < 90.8	GNRI ≥ 90.8	GNRI < 96.9	GNRI ≥ 96.9	GNRI < 92.0	GNRI ≥ 92.0	GNRI < 89.9	GNRI ≥ 89.9	GNRI < 89.5	GNRI ≥ 89.5	GNRI < 94.9	GNRI ≥ 94.9
	*n* = 3311	*n* = 31,622	*n* = 6000	*n* = 15,324	*n* = 1261	*n* = 7445	*n* = 794	*n* = 4109	*n* = 1314	*n* = 19,220	*n* = 3780	*n* = 10,619
Number of deaths	1935	10,564	1713	2850	800	3561	686	2889	879	6808	1733	3079
Person-year	12,065	144,282	27,472	74,315	4612	32,812	2101	15,035	4422	85,900	15,617	50,407
Crude rate * (/1000 person-year)	160.4	73.2	62.4	38.4	173.5	108.5	326.6	192.2	198.8	79.3	111.0	61.1

Abbreviations: GNRI, geriatric nutrition risk index. * The difference in crude mortality is significant in all subgroups (*p* < 0.001).

**Table 3 nutrients-15-03831-t003:** Cox-proportional hazard ratios (95% CI) for all-cause mortality of patients with GRNI in 2015.

	GNRI	Univariate	Model 1	Model 2	Model 3
Total	GNRI < 90.8	2.24 (2.14–2.35)	1.73 (1.65–1.82)	1.71 (1.63–1.79)	1.78 (1.69–1.88)
	GNRI ≥ 90.8	ref	ref	ref	ref
Age < 65	GNRI < 96.9	1.64 (1.54–1.74)	1.59 (1.50–1.69)	1.55 (1.46–1.65)	1.65 (1.54–1.76)
	GNRI ≥ 96.9	ref	ref	ref	ref
Age 65–74	GNRI < 92.0	1.64 (1.52–1.77)	1.58 (1.46–1.71)	1.58 (1.46–1.71)	1.65 (1.52–1.80)
	GNRI ≥ 92.0	ref	ref	ref	ref
Age ≥ 75	GNRI < 89.9	1.78 (1.64–1.94)	1.70 (1.56–1.85)	1.67 (1.53–1.81)	1.67 (1.52–1.84)
	GNRI ≥ 89.9	ref	ref	ref	ref
Men	GNRI < 89.5	2.58 (2.41–2.77)	1.86 (1.73–2.00)	1.84 (1.72–1.98)	1.92 (1.78–2.08)
	GNRI ≥ 89.5	ref	ref	ref	ref
Women	GNRI < 94.9	1.84 (1.74–1.96)	1.55 (1.46–1.65)	1.55 (1.46–1.65)	1.59 (1.48–1.70)
	GNRI ≥ 94.9	ref	ref	ref	ref

Abbreviations: GNRI, geriatric nutritional risk index. Multivariate Model 1 included age, sex, dialysis vintage, and GNRI; Model 2 included the variables of Model 1 plus comorbid conditions; Model 3 (full model) included all previous variables plus laboratory findings.

**Table 4 nutrients-15-03831-t004:** Cox models with GNRI as a time-varying factor among the patients with GNRI in 2015 and 2018.

	GNRI	Model 1	Model 2	Model 3
Total	GNRI (continuous)	0.96 (0.96–0.97)	0.96 (0.96–0.97)	0.96 (0.95–0.97)
	GNRI < 90.8 vs. ≥90.8	1.73 (1.56–1.91)	1.74 (1.57–1.92)	1.74 (1.56–1.95)
Age < 65	GNRI (continuous)	0.96 (0.95–0.97)	0.96 (0.95–0.97)	0.96 (0.94–0.97)
	GNRI < 96.9 vs. ≥96.9	1.44 (1.27–1.64)	1.43 (1.26–1.62)	1.50 (1.30–1.73)
Age 65–74	GNRI (continuous)	0.96 (0.95–0.97)	0.96 (0.95–0.97)	0.96 (0.94–0.97)
	GNRI < 92.0 vs. ≥92.0	1.75 (1.49–2.05)	1.76 (1.50–2.07)	1.78 (1.49–2.12)
Age ≥ 75	GNRI (continuous)	0.97 (0.96–0.98)	0.97 (0.96–0.98)	0.97 (0.96–0.98)
	GNRI < 89.9 vs. ≥89.9	1.57 (1.34–1.84)	1.58 (1.34–1.85)	1.62 (1.35–1.93)
Men	GNRI (continuous)	0.96 (0.96–0.97)	0.96 (0.95–0.97)	0.96 (0.95–0.97)
	GNRI < 89.5 vs. ≥89.5	1.84 (1.59–2.13)	1.85 (1.59–2.14)	1.89 (1.60–2.23)
Women	GNRI (continuous)	0.96 (0.95–0.97)	0.96 (0.95–0.97)	0.96 (0.95–0.97)
	GNRI < 94.9 vs. ≥94.9	1.49 (1.32–1.68)	1.49 (1.32–1.68)	1.48 (1.29–1.69)

Abbreviations: GNRI, geriatric nutritional risk index. Multivariate Model 1 included age, sex, dialysis vintage, and GNRI; Model 2 included the variables of Model 1 plus comorbid conditions; Model 3 (full model) included all previous variables plus laboratory findings.

## Data Availability

The data that support the findings of this study are available from HIRA, but restrictions apply to the availability of these data, so they are not publicly available. Data are, however, available from the authors upon request and with permission from HIRA.

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
