# Peer review of "The Geriatric Nutritional Risk Index Is an Optimal Evaluation Parameter for Predicting Mortality in Adult All Ages Hemodialysis Patients: A Korean Population-Based Study"

_nutrients, 2023, doi:10.3390/nu15173831_

Round 1
Reviewer 1 Report
I had an opportunity to review the manuscript entitled “The optimal value for, and changes in, the Geriatric Nutritional Risk Index for predicting mortality in hemodialysis patients: a Korean population-based study”.
Please find my comments and suggestion below.
1. Please described follow-up process more precisely (please explain HIRA, how the time of observation was calculated, censoring etc).
2. I think that multivariable models were performed not multivariate.
3. I suggest described more precisely how the GNRI was analyzed in the methods section. Moreover, why the GINRI (at baseline) was analyzed according to quartile and according to the cutoff point obtained in the previous analysis?
4. “…GNRI categorized according to the cutoff value19 in the 2015 cohort’. Which method was used to find the cutoff point?
5. Please explain why the Harrell’s C-index (concordance index) was not used to assess the goodness of fit of conducted models instead of likelihood ratio.
Author Response
- Please described follow-up process more precisely (please explain HIRA, how the time of observation was calculated, censoring etc).
--> Thank you for your comments. HIRA is a government organization to assess medical services and maintain quality standards by reviewing healthcare claims. Medical providers are required to submit all of inpatient and outpatient claims to HIRA for reimbursement of medical procedure costs covered by NHIS. We added this in the Methods section.
Patients who received a kidney transplant during a follow-up period or who followed until the end of the study (November 30, 2021) were censored. There were 12,499 deaths and 3,512 received kidney transplantation during the study period out of patients in the 2015 cohort. We added this in the Methods and Results section.
- I think that multivariable models were performed not multivariate.
--> We performed multivariate Cox proportional hazard regression analysis to assess the associations between all-cause mortality and GNRI values/changes. Three different multivariate models were used after adjustment. A total of 15 variables were used in the full model.
- I suggest described more precisely how the GNRI was analyzed in the methods section. Moreover, why the GINRI (at baseline) was analyzed according to quartile and according to the cutoff point obtained in the previous analysis?
-->With your recommendation, we revised the methods section. We analyzed baseline GNRI by quartile to show trends of clinical characteristics in the study population.
- “…GNRI categorized according to the cutoff value19 in the 2015 cohort’. Which method was used to find the cutoff point?
-->The GNRI value from 80 to 120 was subdivided by every 0.1. Each value was used as the cutoff point in the Cox models. We revised the method section.
- Please explain why the Harrell’s C-index (concordance index) was not used to assess the goodness of fit of conducted models instead of likelihood ratio.
-->Thank you for your comments. We used likelihood ratio tests to choose the “best” model between two nested models. The nested models mean (1) one including the explanatory variables without the GNRI and (2) the other including the GNRI categorized according to the cutoff value. Meanwhile, concordance index is a "global" index to validate the predictive ability of a survival model. It is a fraction of pairs in the data, where observation with a higher survival time has a higher probability of survival predicted by the model. The index is not calculated for every observation/subject. So, the c-index cannot be interpreted as a risk of a subject. When we analyzed the Harrell’s C-index of the model with cutoff value of 89.5, the concordance index was 0.727. We added the result in the Results section.
Reviewer 2 Report
Dear authors and editor
This is a korean based study with a large number of patients (>16K) by using ther national health insurance system and the record of Geriatric Nutritional Risk Index (GNRI) in 2015 and 2018.
There was a similar study in 2020 at Japan which measuring GNRI to predict mortality in hemodialysis patients (more than 3k), which reveals similar result. (ref 1) So it seems to me like an "metoo" study.
Lower GNRI showed higher risk of mortality is a well established fact, as the authors stated at the second paragraph in the introduction. So the change trend of GNRI , if decreasing, means higher risk is no surprising at all. (table 6)
The main highlight point of this article is the cut-off value for GNRI, which the authors suggest changing from several values provided by other studies (91.2, 92, or 97.7) to 89.5.
But this value, due to the limitation (mentioned by author at the last two paragraph at discussion,) can only present as a tool which is most accurate for korean population at certain period of time window (from 2015 to 2018.) So the importance of this finding is questionable.
The author did not exam whether using other value (like 90 or the upper limit of the 4th quartile 95.2 from table 1) can achieve similar result or not. Figure 2 showed the cut-off value around 90 may has similar likehood ratio as 89.5 does. Maybe a ROC test can help to clarify it.
Overall the scientific soundness of this study is good but the novelty is low. But I appreciate the chanece for this interesting sharing.
minor issue
GNRI is 14.89 with alb (g/dl), then add BW factor. (not 1.489 in page 3)
I think the auhor confuse the unit g/dLwith g/L.
Reference
Yamada S, Yamamoto S, Fukuma S, Nakano T, Tsuruya K, Inaba M. Geriatric Nutritional Risk Index (GNRI) and Creatinine Index Equally Predict the Risk of Mortality in Hemodialysis Patients: J-DOPPS. Sci Rep. 2020 Apr 1;10(1):5756. doi: 10.1038/s41598-020-62720-6. PMID: 32238848; PMCID: PMC7113241.
Author Response
There was a similar study in 2020 at Japan which measuring GNRI to predict mortality in hemodialysis patients (more than 3k), which reveals similar result. (ref 1) So it seems to me like an "metoo" study.
-->Thank you for your comment. There are several studies of association between GNRI and all-cause mortality in HD patients. The aims of a Japanese study in 2020 are to confirm that lower GNRI and Cr index values are associated with increased risk of mortality in HD patients and to determine whether the GNRI and the Cr index are equally valuable in the population. This study examined an optimal cutoff value of the GNRI to predict mortality. Prognostic value of GNRI is well-known, but the suggested GNRI values were different by various clinical settings. We wanted to suggest the cut-value of GNRI.
Lower GNRI showed higher risk of mortality is a well established fact, as the authors stated at the second paragraph in the introduction. So the change trend of GNRI , if decreasing, means higher risk is no surprising at all. (table 6)
-->Thank you for your comment. It is well-known that HD patients have a higher mortality risk than general population even with technical advances. Several factors like old age, diabetes, malnutrition and cardiovascular disease are known as prognostic for mortality. Among them, nutrition status can be monitored and timely interventions contribute to better clinical outcomes in this population. Nutritional status in patients undergoing HD constantly changes and longitudinal monitoring may be useful to improve clinical outcomes. GNRI is a useful nutritional assessment tool and prognostic marker, therefore we investigated the prognostic value of GNRI and GNRI changes.
The main highlight point of this article is the cut-off value for GNRI, which the authors suggest changing from several values provided by other studies (91.2, 92, or 97.7) to 89.5. But this value, due to the limitation (mentioned by author at the last two paragraph at discussion,) can only present as a tool which is most accurate for korean population at certain period of time window (from 2015 to 2018.) So the importance of this finding is questionable.
-->We included HD population of Korea where racial diversity is limited. However, it was a nationwide cohort study. Although we could not include multi-racial subjects, we suggest a cut-value of GNRI and needs of validation based on this value in other countries.
The author did not exam whether using other value (like 90 or the upper limit of the 4th quartile 95.2 from table 1) can achieve similar result or not. Figure 2 showed the cut-off value around 90 may has similar likehood ratio as 89.5 does. Maybe a ROC test can help to clarify it.
-->Thank you for your comments. We used likelihood ratio tests to choose the “best” model between two nested models. The nested models mean (1) one including the explanatory variables without the GNRI and (2) the other including the GNRI categorized according to the cutoff value.
Meanwhile, ROC (Receiver Operating Characteristic) curve helps us visualize the true positive rate or true negative rate of a prediction based on logistic regression model. We performed a ROC test with your recommendation. The sensitivity and specificity were 0.121 and 0.958 in 89.5 of GRNI, and 0.215 and 0.901 in 92.5, respectively.
We also performed Harrell’s C-index of the survival model with the cutoff value. The C-index was 0.727. We added this in the Results section.
minor issue
GNRI is 14.89 with alb (g/dl), then add BW factor. (not 1.489 in page 3)
I think the auhor confuse the unit g/dLwith g/L.
-->We confused the unit in the manuscript and revised that, Thank you
Reviewer 3 Report
This is a very interesting study, in which authors defined the cut-off value of GNRI useful to predict mortality in ESRD patients undergoing hemodialysis. The final cut-off value of 89.5 was shown to be optimal to identify patients at risk of poor outcomes. These studies has some major strength (appropriate sample size and selected population of patients with ESRD). The methodology is adequately prescribed and study results are well displayed. I have some concerns to be addressed:
-Validation of cut-off point: this study is performed in a numerous sample of over 30,000 individuals. The authors found that 89.5 is a good cut-off to discriminate patients with risk of poor outcomes. To validate the results and add more significance and robustness to study findings, it would be useful to:
a) validate internally the cut-off points in a randomly selected sub-sample of the study cohort.
b) perform sensitivity analyses by excluding patients with very good GNRI. My question is: "Did GNRI carry a graded increase in mortality or not? I mean, which was the risk of death of patients with GNRI > 89.5 within the last quartile? Or between patients with GNRI < 99.2?
-Discrimination ability of GNRI should be reported, by displaying ROC curves. Indeed, sensitivity and specificity of different cut-off values (3 + the chosen cut-off of 89.5) is needed to be reported.
-Which is the prognostic accuracy of GNRI < 89.5 in terms of C index (or Harrell's C?).
Only some minor English spells need to be corrected.
Author Response
-Validation of cut-off point: this study is performed in a numerous sample of over 30,000 individuals. The authors found that 89.5 is a good cut-off to discriminate patients with risk of poor outcomes. To validate the results and add more significance and robustness to study findings, it would be useful to:
a) validate internally the cut-off points in a randomly selected sub-sample of the study cohort.
-->Thank you for your comment. We selected 20 percent of the study population as a sub-sample and calculated the likelihood ratio test statistic in two fully adjusted Cox models. We performed this validation analysis one hundred times. Minimum and maximum values of the cutoff in the sub-sample were 84.2 and 98.2, respectively. 90.0 was the most frequent value, similar to our result.
b) perform sensitivity analyses by excluding patients with very good GNRI. My question is: "Did GNRI carry a graded increase in mortality or not? I mean, which was the risk of death of patients with GNRI > 89.5 within the last quartile? Or between patients with GNRI < 99.2?
-->We performed sensitivity analyses by excluding patients with GNRI ≥ 102.8 and multivariate Cox regression analyses (Model 3). Mortality risk in patients with GNRI <89.5 was significantly higher than ones with 89.5 ≤ GNRI < 102.7 (HR 1.85, 95% CI 1.74 – 1.96) and with 89.5 ≤ GNRI < 99.2 (HR 1.73, 95% CI 1.63 – 1.84). We added this in the Results section.
-Discrimination ability of GNRI should be reported, by displaying ROC curves. Indeed, sensitivity and specificity of different cut-off values (3 + the chosen cut-off of 89.5) is needed to be reported. Which is the prognostic accuracy of GNRI < 89.5 in terms of C index (or Harrell's C?).
-->Thank you for your comments. We used likelihood ratio tests to choose the “best” model between two nested models. The nested models mean (1) one including the explanatory variables without the GNRI and (2) the other including the GNRI categorized according to the cutoff value.
Meanwhile, ROC (Receiver Operating Characteristic) curve helps us visualize the true positive rate or true negative rate of a prediction based on logistic regression model. We performed a ROC test with your recommendation. The sensitivity and specificity were 0.121 and 0.958 in 89.5 of GRNI, and 0.215 and 0.901 in 92.5, respectively.
We also performed Harrell’s C-index of the survival model with the cutoff value. The C-index was 0.727. We added this in the Results section.
Round 2
Reviewer 2 Report
No further question.
Minor editing of English language required
Author Response
Thank you for your comments and recommendation.